# Anticancer Attributes of Cantharidin: Involved Molecular Mechanisms and Pathways

**DOI:** 10.3390/molecules25143279

**Published:** 2020-07-19

**Authors:** Faiza Naz, Yixin Wu, Nan Zhang, Zhao Yang, Changyuan Yu

**Affiliations:** College of Life Science and Technology, Beijing University of Chemical Technology, Beijing 100029, China; faizanaz211@gmail.com (F.N.); wyx19960412@163.com (Y.W.); 17801170951@163.com (N.Z.)

**Keywords:** cantharidin, blister beetles, anticancer, molecular mechanism, cancer

## Abstract

Cancer is a preeminent threat to the human race, causing millions of deaths each year on the Earth. Traditionally, natural compounds are deemed promising agents for cancer treatment. Cantharidin (CTD)—a terpenoid isolated from blister beetles—has been used extensively in traditional Chinese medicines for healing various maladies and cancer. CTD has been proven to be protein phosphatase 2A (PP2A) and heat shock transcription factor 1 (HSF-1) inhibitor, which can be potential targets for its anticancer activity. Albeit, it harbors some toxicities, its immense anticancer potential cannot be overlooked, as the cancer-specific delivery of CTD could help to rescue its lethal effects. Furthermore, several derivatives have been designed to weaken its toxicity. In light of extensive research, the antitumor activity of CTD is evident in both in vitro as well as in vivo cancer models. CTD has also proven efficacious in combination with chemotherapy and radiotherapy and it can also target some drug-resistant cancer cells. This mini-review endeavors to interpret and summarize recent information about CTD anticancer potential and underlying molecular mechanisms. The pertinent anticancer strength of CTD could be employed to develop an effective anticarcinogenic drug.

## 1. Introduction

Cancer is perceived as a deleterious health disorder throughout the world [1]. Although it is regarded as the second leading cause of world mortality, according to recent research, it is ranked first for fatalities in high-income countries [2]. Pursuant to the American Cancer Society, it is predicted that it will cause around 1.8 million new cancer patients and 606,520 deaths in the US alone by 2020 [3]. Cancer is a heterogeneous disorder that is comprised of multifarious cells, linked together to sustain abnormal growth and proliferation of the afflicted cells. The major hallmarks of the cancer include enhanced proliferative signals, increased angiogenesis, metastatic invasion, aneuploidy and immune impairment which results in immortality of the affected cells and ultimately death of the patient [4]. This disease remained one of the most challenging disorders to cure, mainly due to the higher heterogeneity of the cells within the cancer microenvironment. Cancer cells isolated even from the same site possess a high degree of variations [5]. With the passage of time, scientists have proposed different strategies to cure this life-threatening disorder. Some approaches include conventional medicines, chemotherapy, radiotherapy, surgery targeted therapy and immunotherapy [6,7]. Regrettably, these impediments harbor some side effects such as dermatological toxicities, vomiting, nausea, anemia, loss of appetite, fatigue, hypersensitivity and neurotoxicity which result in deprived organ functionality and compromised quality of patient’s life [8,9].

Considering these dilemmas, there is a great need to investigate naturally occurring bioactive compounds to fight this fatal disorder as natural toxins possess certain therapeutic effects on various diseases and are the valuable repository for modern drug discovery. For example, Artemisinin from *Artemisia annua* is a notable antimalarial and antiviral compound against multidrug-resistant malarial strains, HBV, HCV and HCMV [10,11]. Resveratrol, another natural phenolic compound in red grapes and berries, can be utilized to improve prognosis in patients with inflammatory bowel disease [12]. Similar to plants, some natural products from insects have also been reported as potential therapeutic agents to treat various medical disorders as venom from *Apis dorsata*, *Nasonia vitripennis* and *Bracon hebetor* can be utilized as anti-inflammatory agents in mammalian cell lines or mice models [13]. One of such efficacious bio toxin is cantharidin (CTD), extracted from male blister beetles of meloid family (*Mylabris phalerata*, *Mylabris cichorii*) for various medical purposes (Figure 1) [14,15]. The anticarcinogenic efficacy of this terpenoid toxin was first highlighted by Chen et al. around four decades back (1980) [16]. After it, further researches were conducted to elucidate its antitumor effects and finally, it has been revealed that its anticancer activity is mainly due to the inhibition of protein phosphatase 1 (PP1) and protein phosphatase type 2A (PP2A) [17]. Moreover, it has also been proven that CTD can inhibit expression level of HSP70 and BAG3 proteins by preventing heat shock transcription factor 1 (HSF-1) binding to its promoter [18,19]. Following extensive research, it was found that CTD could suppress liver, pancreatic, colon, bladder and breast cancer. Moreover, its antitumor effects against oral carcinoma and leukemia has also been reported [14]. This review aims to highlight molecular mechanisms behind CTD anticarcinogenic potential and summarizes recent information regarding its antineoplastic effects in numerous cancer cell lines.

## 2. Natural Sources of CTD and Its Synthetic Derivatives

Insects of the meloid family beget CTD as a defensive vesicant against predators. It also plays a significant role in mating as male beetles produce it as sexual attractant which is later transferred to female beetles during copulation [21,22]. Although it is only produced by male beetles, its concentration remains 5–6% higher in some female species [23]. Insects of meloid species are found all over the world aside from New Zealand and Antarctic [24]. This family harbors around 3000 species and 125 genera [25], which are mainly inhabited to temperate and barren zones and in subtropical and tropical savannas [26].

Although CTD is toxic in nature and it harbors some toxic effects such as renal toxicity, dysphagia and liver congestion, but its anticancer potential can’t be abandoned as it possesses many pertinent anticancer effects on cancerous cells like PP1 and PP2A inhibition, apoptosis induction and protein synthesis alteration [27]. In order to avoid lethal side effects of CTD, it is necessary that its delivery should be cancer specific. Furthermore, ethanolamine (ETA) can serve as a powerful antidote to rescues CTD cytotoxicity as it particularly alters phosphatidylethanolamine (PE)-associated functions [28] Moreover, thousands of its derivatives have been synthesized to overcome its toxicity d and some of these have tremendous anticancer abilities [14]. Certain important derivatives of CTD are norcantharidin, norcantharimide, cantharidinamides, sodium cantharidate, anhydride-modified derivatives and N-hydroxycantharidimide (Figure 2) [14,27].

## 3. Biologic Features of CTD

Historically, blister beetle’s dried bodies are utilized in Chinese traditional medicines to treat different medical disorders [30]. It was used to cure different skin related ailments such as warts and Molluscum contagiosum, a cutaneous or mucosal viral infection [27]. Ancient Asian people used it to heal tuberculosis scrofuloderma, ulcers, chronic constipation, and venomous worms. Moreover, it was used as abortifacient and aphrodisiac agent [30,31]. Recently, it has been highlighted that CTD can treat genital warts caused by condyloma acuminatum [32]; (M) Yahya et al. reported that it can induce apoptosis in *leishmania major*, a parasite that infects macrophages and dendritic cells of the immune system and causes cutaneous leishmaniasis [33]. Furthermore, Douglas et al. revealed its therapeutic potential against a diverse group of organisms such as protozoa (*Trichomonas vaginalis*), insects (*Rhopalosiphum padi* and *Myzus persicae*), tick (*Hyalomma lusitanicum*) and a plant-parasitic nematode (*Meloidogyne javanica*). Among these, *trichomonas vaginalis* is a frequent, sexually transmitted parasitic infection caused by protozoa *Trichomonas vaginalis*. Their work enlightens that CTD can strongly inhibit *Trichomonas vaginalis* by 50% [34].

## 4. Anticancer Attributes of CTD

### 4.1. Repression of Cancerous Cell Growth & Proliferation

Cytotoxic drugs are accustomed chemotherapeutic agents to treat cancer. These obstruct the growth and proliferation of cancer cells by triggering DNA damage [12]. The mechanism of CTD impelled DNA damage in carcinoma cells was elucidated by Thomas E et al., in 2005. Their work illustrated that CTD can generate both double-strand and single-strand DNA breaks in a leukemia cell line (CCRF-CEM) [14]. Afterward, the cytotoxic effects of CTD on a wide range of human cancer cell lines were examined by different researchers including skin cancer A431 [35], A375.S2 [36], bladder cancer T24, RT4 [37], non-small cell lung cancer (NSCLC) NCI-H460 [38,39], A549 [40], H358 [41], colorectal cancer colo 205 [42], hepatocellular carcinoma HepG2, Hep3B, gastric cancer BGC823, MGC803 [43], cholangiocarcinoma QBC939 [44], breast cancer cell MCF-7 [45], pancreatic cancer PANC-1, CFPAC-1 [46,47], oral carcinoma SAS, SCC-4, CAL-27 [48], TCA8113 [49], UMSSC [50] and ovarian cancer HO-8910PM [51], which intimate that it can effectively halt growth and proliferation of different human cancers (Figure 3). Imatinib resistance is one of the great challenges for chronic myeloid leukemia (CML) therapy as 20–30% of patients remain resistant to it. CTD can efficaciously inhibit the growth of both CML K562 and CML resistant cells k562R by downregulating BCR-ABL transcription level [52]. Moreover, CTD can induce growth inhibition in triple-negative breast cancer cell lines, MDA-MB-231 [45,53] and MDA-MB-468, which is the most difficult breast cancer type to cure due to resistance to the already established breast cancer therapies, i.e., endocrine therapies and HER-2 targeted therapy [54]. Taken together, these data represents that CTD is a puissant cytotoxic agent that can suppress carcinoma cells growth and proliferation time and dose-dependently.

### 4.2. Induction of Cancerous Cell Apoptosis

Apoptosis is stipulated as a programmed means of cell death, in which a chain of intracellular events leads to the elimination of abnormal or malignant cells. This process is either triggered by extrinsic cellular signals such as stimulation of cell death receptors by extracellular ligands or by intrinsic cellular pathways which are mitochondria dependent [55]. Cancer cells utilize a variety of molecular mechanisms to escape apoptosis which not only facilitates abnormal growth and proliferation of the malignant cells, but also makes it resistant to anticancer therapies [55]. Hence, the identification of apoptosis accelerating therapeutic agents is a novel strategy to cure cancer [56]. Recently, an extensive range of studies have been conducted and it has been documented that CTD can trigger apoptosis in a variety of cancer cell lines through both extrinsic and intrinsic apoptosis pathways as summarized in Table 1 (Figure 4).

#### 4.2.1. Apoptosis: Extrinsic Pathway

The extrinsic pathway of apoptosis is mediated upon binding of an extracellular ligand with their corresponding death receptor [55,73]. Upon binding with ligand this death domain transmits signals to the cell’s cytosol to structure a death-inducing signaling complex (DISC) [56,73]. Once DISC is structured, it causes autocatalytic activation of procaspase-8 [73], which in turn activates apoptotic effector caspase-3, -6 and -7 resulting in cellular apoptosis through extrinsic pathway [55,56]. CTD exerts antitumor effects on the human pancreatic cancer cell line through the extrinsic apoptosis pathway by elevating the expression level of TNF-α, TRAIL-1, TRAIL-2 [60,62]. On the other hand, in the human skin cancer cell line, it was found to upregulate the expression of death associated receptors including DR4, DR5 and TRAIL and enhanced activity of caspase-8, -9 and -3 [35]. In colorectal cancer cells, CTD treatment resulted in increased Fas/CD95 expression level [42]. It has also been reported to increase death-related genes expression including DR5, PUMA, BTG2, NOXA, GADD45 and TRB3 in human oral squamous sarcoma cell line [50]. Other studies represent that it can induce antineoplastic activities through upregulating cleaved caspase-8 [39,42,60,62,63] and caspase-3 activation level [39,42,63], which are mediators of the extrinsic apoptosis pathway. Therefore, it can be stated that CTD may induce apoptosis in neoplastic cells through the extrinsic apoptosis pathway.

#### 4.2.2. Apoptosis: Intrinsic Pathway

Intrinsic pathway is mitochondria-associated and regulated by B cell lymphoma 2 proteins (Bcl-2). These proteins act as an apoptotic switch, as they conduct regulation of the mitochondrial outer membrane permeabilization (MOMP) [55]. Upon MOMP, mitochondria release distinct death modulators, namely Cyt C, apoptosis inducing factor (AIF) and endonuclease G (Endo G) into the cytosol to initiate mitochondrial associated caspase-dependent or caspase-independent pathway of apoptosis [74].

CTD can accelerate apoptosis in different cancerous cell lines via intrinsic apoptosis pathway. In human lung cancer cell line A540, CTD induced apoptosis via increasing expression level of Bax protein (<2.5-fold) and active caspase-3 (<2.2-fold), while reducing Bcl-2 translation (<0.4-fold) [40]. Whereas, apoptosis of H460 cell line incubated with CTD was found to be related with decreased mitochondrial transmembrane potential (∆Ψm), enhanced reactive oxygen species (ROS) and Ca^2+^ production and increased expression level of Cyc C and Bax, but reduced expression level of Bcl-XL [39]. In triple-negative breast cancer cell line CTD induced apoptosis by enhancing expression level of Bax, cleaved caspase-3 and PARP [54]. In osteosarcoma cells, MNNG/HOS and MG-63, CTD also carried out apoptosis through a mitochondrial-dependent pathway with upregulating Bcl-2, p-Cdc2 and p-AKT [69]. Evidence suggests that CTD exerts cytotoxic effects on pancreatic cancer cells by JNK-dependent pathway, as CTD treatment suppressed pancreatic cancer cell proliferation via stimulating caspase-8 and caspase-9, upregulating expression level of Bad, Bid and Bak, while downregulating Bcl-2 [60]. CTD remarkably reduced tongue squamous cell carcinoma growth in a dose and time-dependent manner, which was linked with the weakened expression level of miR-214, enhanced expression level of p53 and repression of Bcl-2/Bax signaling pathway [49]. On the other hand, in other tongue squamous carcinoma cell lines SAS, CAL-27, SSC-4, it extended the apoptosis-associated signals of caspase-9, -7 and -3, reduced ∆Ψm and Bcl-2 expression level, while enhanced Bax, Bak, Bid and Cyt C release [48]. Similarly, it caused a reduction in mitochondrial polarization, increased caspase-3, -9 and PARP cleavage in some other oral squamous sarcoma cell lines [50]. Apoptosis of bladder and skin cancer cell lines treated with CTD was found to be associated with the alleviated level of caspase-3, -8, -9 activity, increased ∆Ψm, enhanced Ca^2+^ and ROS level to release Cyt C, whereas reduced Bcl-2 and increased Bax and PARP protein expression [35,63]. In colon cancer cells it also accelerated mitochondrial-mediated apoptosis by reducing ∆Ψm to release Cyc C and Apaf-1. In these cells increased ROS and Bax production was also observed [42]. CTD can also induce caspase-independent pathway of apoptosis by releasing AIF and Endo G in human skin, lung and bladder cancer cell lines [35,39].

Endoplasmic reticulum (ER) also performs a crucial role in initiating and regulating cellular apoptosis [75]. CTD can induce apoptosis in bladder cancer cells with the aid of calcium/PKC regulated ER stress pathway, which was related with enhanced phospho-eIF2α and Grp78 expression, enhanced calpain activity and reduced expression of procaspase-12 [37]; (Y) Xi et al. regarded ER stress and UPR initiation as a fundamental mechanism for the anticancer activity of CTD. Treatment of oral squamous carcinoma cells with CTD resulted in increased ER stress-associated signals and UPR associated proteins (elF2α phosphorylation, XBP1 splicing, accumulation of CHOP and ATF4), which lead to mitochondrial-mediated apoptosis [50]. The apoptotic effects of CTD were also found to be linked with ER stress-mediated pathway in skin cancer cells as it accomplished the upregulation of PERK, IRE1α, GRP78, GADD153, calpain-1, calpain-2 and cleaved ATF6β [35]. Furthermore, in the lung cancer cell line, it promoted ATF6β, IRE1α, IRE1β, GRP78, caspase-4, calpain-2 and XBP-1 expression to mediate apoptosis via ER stress associated pathway [39].

### 4.3. Effect of CTD on Cancerous Cell DNA Damaging and Repair Associated Proteins

DNA repair mechanism is deemed intrinsic for the viability of either normal or cancer cells as it can abridge or fix DNA damage [76]. Therefore, DNA repair halting drugs are considered auspicious to demolish tumors [77]. In numerous cancer cells, CTD has been reported to alter genomic integrity, as it can produce DNA fragmentation in lung H460, colon colo 205, and skin A431 cells [35,39,42]. Treating NCI-H460 cells with CTD resulted in alteration of the DNA repair and damage associated genes expression, as it reduced BRCA-1, ATM, 14-3-3σ, MGMT, DNA-PK and MDC1 proteins expression. On the other hand, expression levels and cytoplasm to nucleus translocation of phosphorylated p53, MDC1 and phosphorylated H2A.X was increased [76]. In another study, cDNA microarray analysis revealed that CTD caused up-regulation of GADD45A (2.60-fold) and DNIT3 (2.26-fold) and down-regulation of DdiT4 (3.14-fold), which are DNA damage associated genes [38]. CTD caused sensitization of pancreatic cancer cells towards radiotherapy by elevating DNA damage and suppressing DNA repair associated genes namely UBE2T, RM1, RPA1, XRCC1, GTF2HH5, RAD51B, RAD50, RAD51B, PRKDC, LIG1, FANC1, DMC1, POLD3 and FAAP100 through JNK, ERK, p38, PKC and NF-κB pathways [78]. In CML cells CTD has been found to trigger DNA damage via elevating γH2AX, which is an indicator of DNA double-strand breaks [52]. Likewise, in osteosarcoma cells, it brought DNA damage and condensation with increasing active PARP, p-ATR, p-ATM and DNA-PK [68]. Zhang et al., reported inhibitory effects of CTD on promyeloid leukemia cells. In response to CTD treatment, HL-60 cells showed reduction in DNA replication and repair associated genes including DNA polymerase delta, FANCG, ERcc2, hMSH6 and RuvB-like DNA helicaseTIP49b [79] The DNA damaging response of CTD was also found in bladder cancerous cells, as it produced DNA comet tail and DNA condensation upon CTD treatment. This effect was accomplished by reducing PARP, BRCA-1, DNA-PK, MDC1, MGMT, ATR and phosphohistone H2A.X level and increasing p-p53 accumulation [80]. Furthermore, the cDNA microarray showed that it increased DDIT3 gene expression (4.75-fold) [81]. These data typify CTD as a promising agent for inducing DNA damage and suppressing its repair mechanism in tumor cells as summarized in Table 1 (Figure 4).

### 4.4. Induction of Cancerous Cell Cycle Arrest

Abnormal proliferation of the malignant cells lead to divergent activity of several cell cycle regulatory genes, therefore, targeting expression of cell cycle regulatory genes to arrest cells cycle—a state when the cell is no longer capable to duplicate and divide [82]—is considered an alluring approach to halt uncontrolled growth of the cancerous cells [83]. In higher organisms, the cell cycle is a highly controlled event, regulated by various mechanisms. A group of connate proteins, namely cyclins, cyclin-dependent kinases (CDKs) and CDKs inhibitors (CDKi) ensure appropriate progression of cell cycle within a cell [84]. *p*21 and *p*53 are examples of the negative regulators of the cell cycle [56,85]. CTD arrested bladder carcinoma cells at G_0_/G_1_ phase by upregulating p21 and p53 gene translation level and downregulating Cyclin E and CDC25C [63]. The percentage of control vs. CTD treated cells in G_0_/G_1_ phase was found to be 43.31% and 52.14%, respectively [63]. Treatment of CML cells K562 and imatinib-resistant cells K562R with CTD resulted in mitotic arrest, which was mediated by cyclin B1/Cdc2 complex activation and cyclin D1 downregulation. After 24 h of treatment, 19.2–24.5% of K562 cells (control 1.6%) and 10.8–13% of K562R cells (control 1.6%) were arrested in mitotic phase [52]. CTD significantly reduced skin cancer cell growth via arresting cells at G_0_/G_1_ phase with elevating p21 and lowering Cyclin D, Cyclin E and CDK6 expression level [35]. CTD arrested colon cancer cells in G_2_/M phase via halting CDK1 activity [42]. On the other hand, it vitalized the APC complex through PP2A inhibition and CDK1 downregulation to arrest pancreatic cells in G_2_/M phase [60]. CTD also induced G_2_/M phase arrest in osteosarcoma cells with increasing CHK1, phosphorylated p53 and WEE1 expression level, while decreasing CDC25C and CDK1 expression [68]. Moreover, it was able to induce G_2_/M phase in hepatocellular carcinoma derived stem cells, CD133^+^ [71], breast cancerous cells, MDA-MB-231 [53], colorectal cancerous cells, HCT-116 [18] and in renal cancerous cells, ACHN and Caki-1 RCC [67]. Therefore, it is certain that CTD can arrest the cancerous cell cycle by regulating various cell cycle-associated proteins expression levels as summarized in Table 1 (Figure 4).

### 4.5. Inhibition of Cancer Cell Metastasis

Neoplastic metastasis typifies an advanced phase of malignancy, thus, resulting in fatality in around 90% of cancer cases [56,86]. Metastasis is known to be a multiphase mechanism that involves invasion and migration of the tumor cells to neighboring tissues or organs [35]. Cancer metastasis is considered as a major obstacle in the effective treatment of the cancer patients [56]. Therefore, anti-metastasis drug development is getting more attention [65]. Cancer cells cause degradation of the extracellular matrix (ECM) to invade normal tissues. Matrix metalloproteinases (MMPs) play a pivotal role in ECM degradation [36]. To date, more than 20 MMPs are recognized and among these, MMP-2 and MMP-9 are considered intrinsic for cancer metastasis [66].

CTD remarkably averts the metastatic potential of different cancer cells by deregulating various metastasis-associated proteins expression. In gastric cancer cells, CTD inhibited migration and invasion by PI3 K/AKT signaling pathway that was mediated by down-regulating CCAT1 [43]. CTD concentration-dependently halted the adhesion, migration and invasion of bladder cancer cells TSGH-8301 by altering p38 and JNK1/2 MAPK signaling pathway to down-regulate MMP-2 and MMP-9 mRNA, protein level and enzymatic activity [64]. In NCI-H460 cells, CTD repressed migration, invasion and adhesion by arresting MAPK signaling pathway via reducing NF- ĸB p56 and AKT, leading to UPA protein decreased expression and decreased enzymatic activity of MMP-2 and MMP-9 [65]. In another lung cancer cell line A549, the anti-metastatic effect of CTD remained different as it only inhibited the gelatinous efficacy of MMP-2, but not MMP-9, while the expression level of either MMP-2 or MMP-9 had not changed. This effect was linked with PI3 K/AKT signaling pathway repression, but not MAPK signaling pathway [66]. Similar to another study, CTD treatment caused inhibition of the A549 cells migration via PIk3/Akt/mTOR pathway repression [40]. In ovarian cancer cells, CTD instigated downregulation of VEGF and NF-ĸB p65 subunit to repress invasion, adhesion and migration [51]. CTD substantially inhibited invasion and migration ability of A375.S2 cells by impeding the expression level of MAPK signaling pathway proteins including p38, JNK and ERK via depleting NF-ĸB and AKT. CTD also caused suppression of MMP-2, MMP-9, FAK, PI3 K and ERK1/2 translation level in these cells [36]. The metastasis ability of triple-negative breast cancer cell line was found to be linked with MAPK pathway modulation/inactivation as CTD treatment caused decrease phosphorylation of MEK, MAPK, JNK, p38 and ERK. Furthermore, reduction in MMP-2 and MMP-9 expression level promoted CTD-directed suppression of MDA-MB-231 cells migration and invasion [53]. On the other hand, in MCF-7 cells, it suppressed growth and adhesion with downregulation of α2 integrin—a cancer cells surface adhesion molecule—by protein kinase C pathway [45]. Recently, Pan et al. revealed that CTD can exert anti-metastatic effects on breast cancer cells by transforming aerobic glycolysis to oxidation. Divulging this mechanism, they highlighted that CTD can inhibit pyruvate kinase (PK), which causes blockage in pyruvate kinase M2 (PKM2) translocation to the nucleus. This process leads to the downregulation of GLUT1/PKM2 loop which is essential for glucose transport and glycolytic metabolism [17]. CTD attenuated proliferation and migration of pancreatic cancer cells as a consequence of β-catenin-directed repression of Wnt/β-catenin signaling pathway [59]. Moreover, it post-transcriptionally degraded MMP-2 mRNA with JNK, NF-ĸB, ERK, PKC and β-catenin pathways to cause hindrance in the invasion ability of pancreatic cancer cells [58]. CTD also exerted anti-metastatic effects on cholangiocarcinoma cells as in QBC939 cell line, it caused activation of IKKα/IκBα/NF-κB pathway which resulted in MMP-2 and MMP-9 deactivation. Normally, NF-κB p65 positively regulated MMP-2 and MMP-9 protein expression, while in QBC939 cells NF-κB p65 exerted negative effects on these metalloproteinases [70]. Hence, considering these findings it can be stated that CTD is a potent anti-metastatic agent in different cancer cells.

### 4.6. Induction of Cancerous Cell’s Autophagy

Autophagy is defined as an evolutionary conserved catabolic process that constrains cellular hemostasis through self-degrading defective or unwanted cellular organelles that are self-destroyed [87]. It depicts a significant role in multiple biologic diseases such as cancer [56]. Autophagosome formation is a characteristic feature of autophagy and microtubule-associated protein 1 light chain 3 (LC3) is a substantial autophagy associated molecule that assists autophagosome formation [88]. Beclin-1 is another important molecule in initiating autophagy, as it converts LC3-I to LC3-II [87,89]. Lately, CTD was found to instigate autophagy in A549 cells by upregulating Beclin-1 and LC3-I/LC3-II along with downregulating p62 protein expression level. This response was consorted with PI3 K/AKT/mTOR signaling pathway repression, as phosphorylated AKT, phosphorylated mTOR and phosphorylated p70-S6 K levels were greatly decreased [40]. CTD arrested cell cycle at G_2_/M phase in breast, lung and pancreatic carcinoma cells by autophagy associated upregulation of p21 protein. DsRed-LC3 reporter was used to confirm autophagy in these cells, which showed that there was an increase in LC3 punctate formation in CTD treated cells. In these cells, CTD caused JNK mediated CDK1 downregulation, however, p21 elevated level was independent of the JNK pathway, as pretreatment of the CTD treated cells with JNK inhibitor (SP600125) caused no effect on the p21 upregulation. Additionally, autophagy suppression by 3-Methyladenine attenuated p21 increased level. These findings indicate that CTD mediated cytotoxicity and cell cycle arrest through JNK/Sp-1 based repression of CDK1 and autophagy associated elevation of p21 [72]. Autophagy performs a dual role in cancer regulation, as its pro-survival role positively accelerates several hallmarks of cancer [87]. In triple-negative breast cancer cells, CTD was found to restrain pro-survival autophagy. This effect was accomplished via suppression of LC3-I to LC3-II conversion and autophagosome formation through Beclin-1 downregulation [53]. These findings indicate that CTD can effectively induce autophagy via modulating several autophagy associated genes as summarized in Table 1 (Figure 4).

### 4.7. Cytotoxic Effects of CTD in Xenograft Mice Model

In-vivo cytotoxic competence of CTD has also been validated and it was sighted that CTD markedly reduced tumor size in various tumor models. Revealing in vivo cytotoxic effects of CTD in the skin cancer mouse xenograft model, Chi-Chuan Li et al. determined that 0.2 and 1 mg/kg CTD significantly reduced tumor size in S180 tumor-bearing host [35]. On the other hand, in the T24 tumor mice model after 21 days of treatment with 0.5 mg/kg of CTD, there was a 71% reduction in tumor size [37]. Treating triple-negative breast cancer xenograft mice model with 20 mg/kg of CTD for four weeks caused a 46.7% reduction in tumor size than control mice, whereas treatment with 40 mg/kg of CTD resulted in reduced tumor size [53]. In another study, BALB/c nude mice bearing triple-negative breast cancer cells MDA-MB-468, MDA-MB-231 and Beclin-1 gene overexpression were used for in vivo study. Mice were intravenously treated with 10 mg/kg CTD after every two days and tumor volume was evaluated after every two weeks. The results showed that CTD was able to inhibit the growth of the TNBC tumor in vivo via inducing apoptosis and inhibiting pro-survival autophagy [54]. Yanhong Pan et al. verified in vivo anti-metastatic strength of CTD. In MDA-MB-231 breast cancer female mice, CTD was found to inhibit cancer cells metastasis to lung and liver, as 0.5 mg/kg CTD remarkably reduced metastasis foci of cancer in liver and lung [17]. Furthermore, reduction in the number of completely formed tubes, and vascular sprouting density and length indicated inhibition of angiogenesis in these mice [17]. On the contrary, in lung, pancreatic, and colorectal cancer cell mice model CTD exerted proangiogenic effects. Upon CTD treatment in xenograft mice, an increase in proangiogenic proteins and a decrease in antiangiogenic proteins have been observed. These findings highlight proangiogenic side effects of CTD in in vivo model [46]. The antitumor effects of CTD in the EAC mice xenograft model were also identified. In this study, cisplatin was used as a positive control. Results revealed that CTD was responsible for tumor destruction through apoptosis, necrosis and autophagy. CTD caused apoptosis by caspase activation and mitochondrial-dependent intrinsic pathway and it inhibited LDH activity to shorten NAD^+^ and ATP levels leading to metabolic stress, which is an essential factor of autophagy. Additionally, CTD treatment increased the life span of EAC mice by 82%. Surprisingly, normal cells possessed less sensitivity to CTD treatment as they had much greater IC_50_ value than EAC cells. Besides this, only a very little amount of CTD (0.5 mg/kg/day) was responsible for in vivo cytotoxic effects than control (2 mg/kg/day) [90]. These experiments explicate in vivo cytotoxic efficiency of CTD, as it significantly reduced tumor size and induced apoptosis in different cancer models.

## 5. CTD in Combined Therapy

With rapidly growing cancer cases, cancer research financial expenses are also burgeoning [91]. Although there is a pressing need to design new anticancer medications with more antineoplastic propensity, but it takes around 15 years for a newly synthesized anticarcinogenic medication to enter the pharmacological industry [91]. Considering this scenario, combination therapy—a treatment method that involves the amalgamation of two or more drugs—is regarded as a cornerstone in oncology [91]. In light of previous studies, it is accepted that CTD can inhibit growth and proliferation of various pancreatic cancer cells, but effusively stimulated PKC can attenuate CTD cytotoxic effects and improves pancreatic cancer cell’s survival. The combination of tamoxifen—a PKC inhibitor—and CTD can potentially overcome this drawback. It is because CTD causes enhanced phosphorylation of PKCα which was suppressed by tamoxifen. Therefore, cotreatment of CTD (PP2A inhibitor) with tamoxifen (PKC inhibitor) can substantially halt tumor surviving effects of pancreatic cancer cells [92]. Moreover, CTD can enhance the cytotoxic efficacy of gemcitabine and erlotinib, two conventional anti-pancreatic cancer pharmacotherapeutics [47]. Gemcitabine and cisplatin (GP) is identified as the first-choice drug for treating non-small cell lung cancer (NSCLC). However, clinical potency of this drug is finite due to severe side effects such as thrombocytopenia, leukemia and anemia. Aidi injection-combination of CTD and astragalus-containing herb can not only suppress the toxic effects of GP chemotherapy, but also improves NSCLC patient’s quality via enhancing tumor immunity [93]. On the other hand, the combination of *Brucea javanica* and CTD also alleviated the side effects of chemotherapy in NSCLC patients [94]. Consistently, CTD and Shenmai injection in conjunction with chemotherapy (epirubicin hydrochloride, cyclophosphamide, docetaxel) reduced side effects in breast cancer patients [95]. The combination of Qinin—CTD sodium injection—with fluoropyrimidine-based chemotherapy enhanced gastric cancer patient’s survival potential [96]. Furthermore, a combination of CTD and radiotherapy has also been studied. Wang et al. reported that CTD can increase the sensitivity of pancreatic cancer cells to radiation therapy. It strengthens the anti-proliferative potential of radiation therapy by arresting cells in G_2_/M phase and enhancing DNA damage [47]. Conversely, CTD in combination with radiotherapy was more efficacious in rejecting lung cancer. In C57BL/6 mice containing Lewis lung cancer, CTD and radiotherapy synergistically inhibited the growth of lung cancer cells by enhancing T-cell infiltration, cytokine production and proliferation than CTD treatment or radiotherapy alone [97]. Even though CTD in combination with chemotherapy is found to be safer for patients, but these in vivo results need to be further investigated in large sample sizes. In addition, the synergistic effects of CTD with some other natural drugs should be determined.

## 6. Conclusions and Future Directions

This review article epitomizes anticancer attributes of CTD in numerous cancer cell lines. Given the above-mentioned information, it is clear that CTD is an eminent anticancer compound that inhibits PP2A and HSF-1. CTD could indicatively repress cancerous cells growth, proliferation and migration. Moreover, it could induce apoptosis, cell cycle arrest and autophagy and can also attenuate various DNA damaging and repair associated proteins in malignant cells. Nevertheless, the effect of CTD on the cancerous cell’s differentiation needs to be explicated. As cancer is recognized as a multifarious ailment that is developed by multitudinous deformities, it is necessary to treat it with a multi-target drug. Interestingly, CTD can affect various cell signaling pathways, but MAPK, Bcl2/Bax, JNK, NF-κB, ERK, PKC, β-catenin, Wnt/β-catenin, PI3/AKT and PIk3/ATK/mTOR are recognized its potential molecular targets. The antitumor potential of CTD has also been proven in mice xenograft models, but there is a need to further verify it in other cancer models. Moreover, in the breast cancer mice model, it was found to be antiangiogenic, but in the lung, pancreatic and colorectal cancer model, it increased neoplastic cell growth as a consequence of elevated angiogenesis. Thus, it is required to verify whether it is pro or antiangiogenic in other cancer cells. CTD has also been found effective in combination with chemotherapy and radiotherapy as it can remarkably reduce chemotherapy aftereffects and sensitizes cells to radiotherapy. However, in clinical patients, its effect requires more verification in a large sample size. Additionally, there is little information available about its synergistic anticancer effects in combination with other natural compounds.

## Figures and Tables

**Figure 1 molecules-25-03279-f001:**
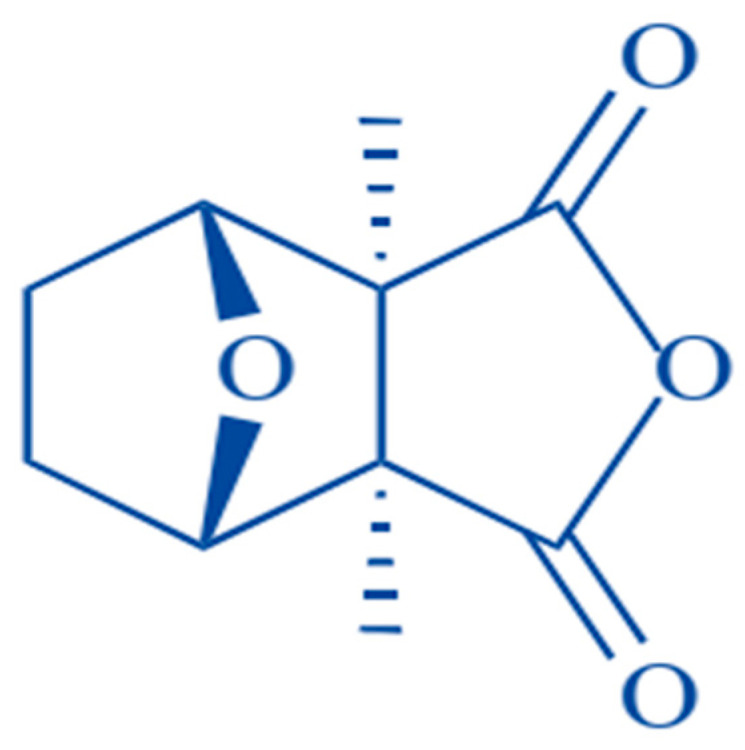
Chemical structure of cantharidin [20].

**Figure 2 molecules-25-03279-f002:**
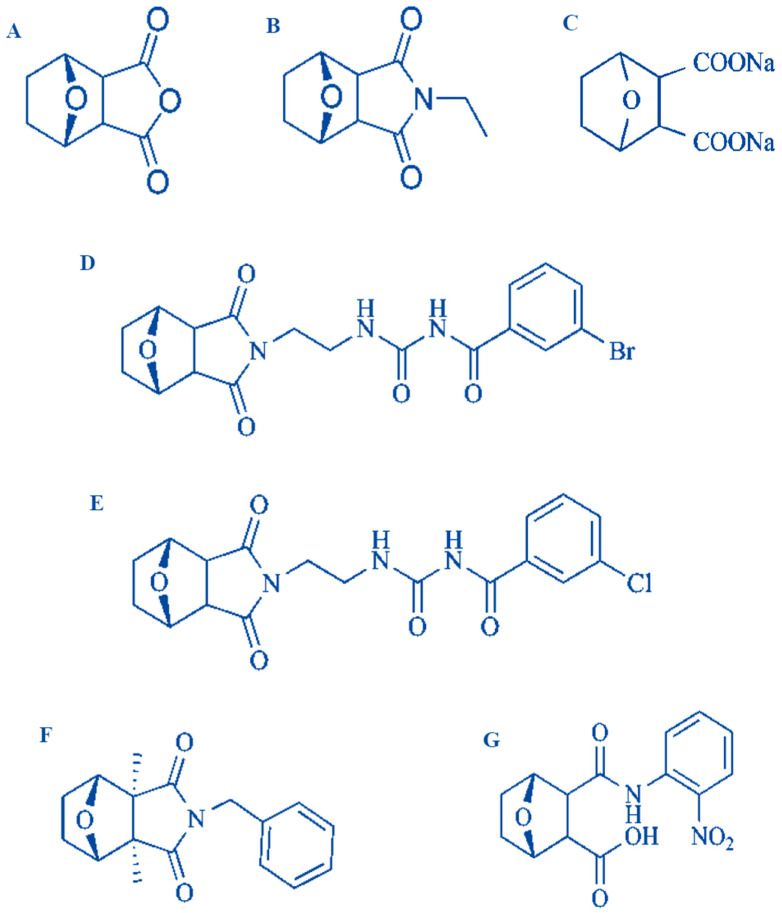
Structure of several derivatives of cantharidin (**A**) Norcantharidin, (**B**) norcantharimide, (**C**) sodium cantharidin, (**D**–**F**) cantharidinamides, (**G**) anhydride-modified derivative of cantharidin [14,27,29].

**Figure 3 molecules-25-03279-f003:**
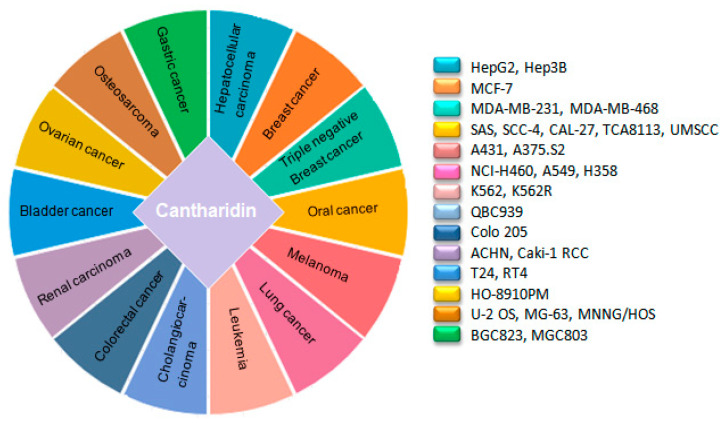
Anticancer profile of cantharidin in different cancer cell lines.

**Figure 4 molecules-25-03279-f004:**
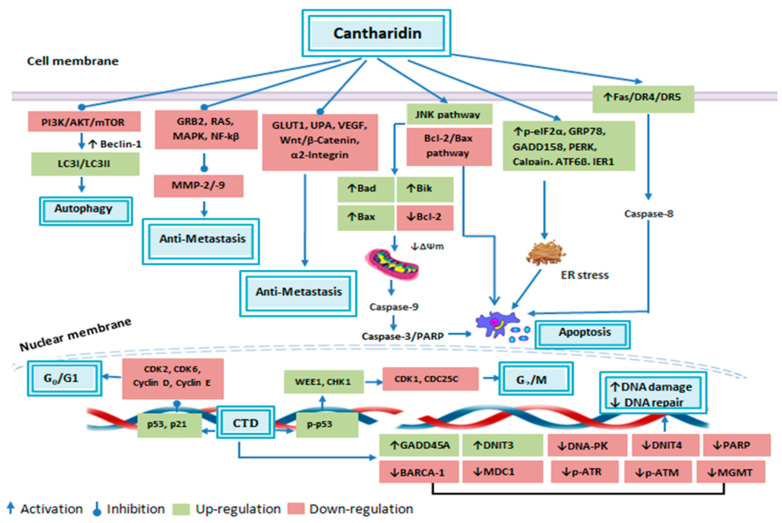
Anticancer attributes of cantharidin and its molecular targets.

**Table 1 molecules-25-03279-t001:** Cantharidin (CTD) anticancer attributes and involved molecular mechanism.

Type of Cancer	Cell Line	Consequence	Molecular Mechanism	Observation Model	Ref
CML	K562, K562R	Growth inhibition, cell cycle arrest, DNA damage	Downregulation of BCR-ABL protein expression	In vitro	[52]
Melanoma	A431	Apoptosis, cell cycle arrest, DNA damage	Caspase-8,-9 & -3 activation, decreased ΔΨm to release Cyc C, Endo G & AIF, increased expression level of DR4, DR5 & TRAIL, $G_{0}$/G1 phase arrest via elevation of p21 while reduction of cyclin D, cyclin E and CDK6 expression level	In vitro, In vivo	[35]
Bladder	T24, RT4	Apoptosis	Induction of apoptosis by calcium/PKC regulated ER stress pathway that involves upregulation of Grp78 & phospho-eIF2a	In vitro, In vivo	[37]
Lung	H460	Apoptosis, cell cycle arrest, DNA damage	Upregulation of DNA damaging genes *DNIT3 & GADD45A* while downregulation of DdiT4, alteration of cell cycle progression genes (*RASA4*, *CCND2*, *CDKL3* upregulation, *CDC42EP3* downregulation) upregulation of apoptosis-associated genes including *CARD6*	In vitro	[38]
Lung	H460	Apoptosis, cell cycle arrest, DNA damage	Increased Ca^2+^ & ROS production, initiation of caspase-3, -8, decreased ΔΨm, increased expression of Cyc C, AIF & Bax & induction of ER stress via upregulation of IRE1σ, IRE1β, GRP78, ATF6α, caspase-4, calpain 2 & XBP1	In vitro	[39]
Melanoma	A375.S2	Growth inhibition, invasion & migration inhibition	Inhibition of migration & invasion via MAPK signaling pathway through NF-ĸB and AKT downregulation resulting in reduction of MMP-2/-9 enzymatic activity and expression level	In vitro	[36]
TNBC	MDA-MB-231, MDA-MB-468	Apoptosis, inhibition of pro-survival autophagy	Inhibition of LC3-I to LC3-II conversion and autophagosome formation through suppression of beclin-1	In vitro, In vivo	[54]
Colorectal	Colo 205	Apoptosis, cell cycle arrest, DNA damage	Elevated activities of caspase-8,-9 & -3, decreased ΔΨm, increased ROS production, stimulation of Cyc C, Fas/CD95 and Bax expression whereas inhibition of Bcl-2 expression, Induction of $G_{2}/M$ phase via CDK1, cyclin A, cyclin B decreased expression and p21 and CHK1 increased expression, induction of apoptosis through increased ROS production & decreased ΔΨm	In vitro	[42]
Breast	MCF-7	Apoptosis, Adhesion inhibition	Adhesion inhibition by α2 integrin downregulation through PKC dependent-pathway		[45]
TNBC	MDA-MB-231	Inhibition of growth, cell cycle arrest, Inhibition of migration & invasion	Suppression of growth & migration via inhibition of MAPK signaling pathway	In vitro, In vivo	[53]
TNBC	MDA-MB-231	Apoptosis	Inhibition of PI3k/Akt & STAT3 signaling pathways by EGF receptor phosphorylation, downregulation of COX-2, Bcl-2 & cyclin $D_{1}$	In vitro	[57]
Pancreatic	PANC-1, CFPAC-1	Inhibition of invasion	Post-transcriptional degradation of MMP2 via NF-κB, PKC, JNK, ERK & β-catenin pathways	In vitro	[58]
Pancreatic	PANC-1	Growth & migration inhibition	Suppression of Wnt/β-catenin pathway through β-catenin phosphorylation & degradation	In vitro	[59]
Pancreatic	PANC-1, CFPAC-1, BxPC-3, Capan-1, Human Pancreatic duct cells, Rat Pancreatic duct cells	Apoptosis, cell cycle arrest	JNK pathway-dependent growth inhibition, Activation of caspase-8 & -9, elevation of TRAILR1, TRAILR2, TNF-α, Bak, Bad & Bik while repression of Bcl-2, $G_{2}$/M phase arrest via p21 upregulation & CDK1 downregulation	In vitro	[60]
Pancreatic	PANC-1	Growth inhibition	Over-activation of JNK pathway	In vitro	[61]
Pancreatic	PANC-1	Apoptosis	NF-κB pathway activation leading to overexpression of TNF-α, TRAIL-1 & TRAIL-2	In vitro	[62]
Tongue squamous cell carcinoma	TCA8113	Apoptosis	Weakened expression of miR-214 leading to p53 upregulation and Bcl-2/Bax pathway downregulation	In vitro	[49]
Oral Squamous Cell Carcinoma	SAS, SSC-4, CAL-27	Apoptosis	JNK-mediated mitochondria & ER stress pathways involving increased expression of caspase-9, -7, & -3, decreased ΔΨm, induction of Cyc C & AIF release, elevated level of Bax, Bak & Bid, reduced expression of Bcl-2, increased expression of p-eIF2 & CHOP, & reduction of pro-caspase-12 expression level		[48]
Bladder	TSGH 8301	Apoptosis, cell cycle arrest, DNA damage	caspase-8, -9, & -3 activation, increased ROS and Ca$2^{+}$generation, decreased ΔΨm, increased AIF & Endo G release, upregulation of Bax & PARP, downregulation of Bcl-2, $G_{0}$/G1 phase arrest in association with decreased cyclin E & Cdc25c, but elevation of p21 & p-p53	In vitro	[63]
Bladder	TSGH 8301	Inhibition of migration, invasion & adhesion	Reduction of MMP-2 & MMP-9 through p38 & JNK1/2 MAPK pathway	In vitro	[64]
Oral squamous cell carcinoma	UMSCC	Apoptosis, DNA damage	Induction of ER stress and activation of UPP	In vitro	[50]
NSCLC	A549	Inhibition of growth, migration & invasion, induction of autophagy	Growth & migration inhibition through induction of autophagy and apoptosis which is consorted with PI3 K/Akt/mTOR pathway repression	In vitro	[40]
NSCLC	NCI-H460	Inhibition of migration, invasion & adhesion	Attenuation of MAPK pathway by reducing NF-ĸB & AKT, leading to down of MMP-2/-9 & UPA	In vitro	[65]
NSCLC	A549	Inhibition of metastasis	Alteration of PIk3/Akt pathway activation resulting in the inhibition of MMP-2 activity	In vitro	[66]
Renal cell carcinoma	ACHN, Caki-1 RCC	Apoptosis, cell cycle arrest	Upregulation of Notch-1 & Jagged1	In vitro	[67]
Osteosarcoma	*U-*2 OS	Apoptosis, cell cycle arrest, DNA damage	Apoptosis induction through both extrinsic & intrinsic pathways, $G_{2}$/M phase arrest via upregulation of CHK-1, WEE-1, CDK-1, p-p53, CDC25C & p21	In vitro	[68]
Osteosarcoma	MG-63 MNNG/HOS	Apoptosis	Increased Bax, PARP whereas reduced Bcl-2 p-Cdc2 & p-Akt expression level	In vitro	[69]
Cholangiocarcinoma	QBC939	Inhibition of migration & invasion	Inhibition of migration and invasion through activation of IKKα/IĸBα/NF-ĸB pathway resulting in suppression of MMP-2 & MMP-9 expression level	In vitro	[70]
Gastric cancer	BGC823, MGC803	Apoptosis, inhibition of metastasis	Suppression of growth & migration by suppressing PI3k/Akt signaling pathway which was mediated by CCAT1 downregulation	In vitro	[43]
Hepatocellular carcinoma	HepG2 CD133^+^	Apoptosis, cell cycle arrest, inhibition of self-renew ability	Halted self-renewable ability by upregulation of β-catenin & cyclin D1, arrested $G_{2}$/M phase by upregulation Myt1, p53, histone H2AX, cyclin A2, Cyclin B1	In vitro	[71]
TNBC	MDA-MB-231	Apoptosis, Inhibition of migration & invasion, induction of angiogenesis	Transformation of aerobic glycolysis to oxidation by breaking GLUT1/PKM glycolytic loop	In vitro, In vivo	[17]
Pancreatic, Breast, NSCLC	PANC-1, T47D, MCF-7, NCI-H292, NCI-H1650	cell cycle arrest	$G_{2}$/M phase arrest via autophagy-dependent upregulation of p21 & JNK/Sp1-dependent downregulation of CDK1	In vitro	[72]

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
