# Peer review of "Anticancer Attributes of Cantharidin: Involved Molecular Mechanisms and Pathways"

_molecules, 2020, doi:10.3390/molecules25143279_

Round 1

Reviewer 1 Report

The manuscript entitled „Molecular Mechanism of Cantharidin Carcinopreventive Potency in Various Cancer Cell Lines” by Faiza Naz et al. is an interesting overview on wide therapeutic possibilities of CTD against cancer. This natural compound, derived from Blister Beetles, has been explored for ages, starting from traditional Chinese medicine. Apart from other therapeutic usage, its anticancer properties seem to be under special attention last decades. In this overview, Authors mentioned results of many studies that confirmed desirable influence of CDT on many cancer cells. An in-depth insight into the various mechanisms of action at the molecular level has been carried out, to give some constructive conclusions at the end.

In my opinion, this overview is enough interesting to be read by wider society of Molecules journal’s readers.

Thus, I recommend this manuscript to be published.

However, several points should be addressed before, as follows:

  1. The chemical side of this manuscript is quite poor. Authors mention several synthetic CDT derivatives (norcantharidin, norcantharimide, cantharidinamides, sodium cantharidate, anhydride- modified derivatives and N-hydroxycantharidimide).

A figure with their structures that can be inspirational for chemical modifications in search for new anticancer drugs should be added in this manuscript

In the same context, it would be interesting to know how popular is the CDT motif in design of new chemical molecules in search for anticancer agents. Authors could provide some data in this field based on recent lines of evidence.

  1. Graphical and language part of this manuscript should be improved

- „in vitro”, „in vivo” „et al.” should be written in italic

- there is no space between reference number in brackets and text

Author Response

Responses to Reviewer #1’s comments

Comment 1: 

The chemical side of this manuscript is quite poor. Authors mention several synthetic CDT derivatives (norcantharidin, norcantharimide, cantharidinamides, sodium cantharidate, anhydride- modified derivatives and N-hydroxycantharidimide).

A figure with their structures that can be inspirational for chemical modifications in search for new anticancer drugs should be added in this manuscript

In the same context, it would be interesting to know how popular is the CDT motif in design of new chemical molecules in search for anticancer agents. Authors could provide some data in this field based on recent lines of evidence.

Response:

Thanks for your suggestion.

A figure describing structure of the Cantharidin derivatives has been provided (Fig 2). This review focuses on the available data about “anticancer attributes of Cantharidin” only, but not its derivatives. Although the structures and anti-cancer effects of Cantharidin derivatives are important for design of new chemical molecules, which could be discussed in another manuscript.

Comment 2:

Graphical and language part of this manuscript should be improved:

- „in vitro”, „in vivo” „et al.” should be written in italic

- There is no space between reference number in brackets and text

Response:

Sorry for these mistakes. These words have been italicized as suggested:

-,,in vitro”  (abstract line 20)

-,,in vivo” (abstract line 20), (page 11 line 340, 348, 350, 351, 358, 366, 367), (page 12 line 396)

-,,et al” (page 2 line 55), (page 3 line 89, 91, 101), (page 5 line 204), (page 6 line 230), (page 10 line 300), (page 11 line 341, 351), (page 12 line 393)

-,,Brucea javanica” (page 12 line 388), (Artemisia annua page 2 line 47), (Apis dorsata, Nasonia vitripennis and Bracon hebetorpage2 line 5), (Mylabris phalerata, Mylabris cichori page 2 line 54), (leishmania major page 3 line 90), (Trichomonas vaginalis page 3 line 93, 94, 95, 96), (Rhopalosiphum padi, Myzus persicae and Hyalomma lusitanicum page 3 line 93), (Meloidogyne javanica page 3 line 94).

-Space between the text and reference number in brackets has also been provided in the whole manuscript.

Reviewer 2 Report

  1. Authors reviewed anticancer effects of cantharidin in various cancer cell lines, mainly relevant to therapeutic effects. This focus is not consistent with the title of this manuscript “Molecular Mechanism of Cantharidin Carcinopreventive Potency…”. Cancer therapy and cancer prevention are different concepts.
  2. As a natural product, cantharidin induces multiple cellular effects which heavily overlap among cancer cell lines. It may not be necessary to list every tested cell lines unless a disease status represented by certain cells is unique to the biological effects of cantharidin.
  3. Several sentences or sub-conclusions are misleading. Here are a few examples: a. “there is a great need to investigate naturally occurring bioactive compounds to fight this fatal disorder as natural toxins possess less pernicious effects.” Natural toxins could be highly detrimental to normal tissue. Authors may fit cantharidin with the concepts of modern drug discovery, emphasizing how the compound may  target the drivers of certain indication.      b. “Revesrsitol (I think the authors mean Resveratrol), another natural phenolic compound in red grapes and berries can be utilized to cure inflammatory bowel disease”. The compound was studied for this disease but can not provide a “cure”.   c. “… but its anticancer potential can’t be abandoned mainly due to its PP2A inhibiting ability.” Many  activities are reported, PP2A inhibition is just one of them.
  4. Analogues of cantharidin could be compared in much detail, including chemical structures, therapeutic and toxic profiles, etc.
  5. Figure 3. It is not clear if the figure is to describe the effects of cantharidin in cytoplasm and in nucleus, as cell membrane and nucleus were drawn. However, a lot of affected proteins in the cytosolic area of the picture exert their functions in the nucleus.
  6. Many misspellings throughout the manuscript need to be corrected.

Author Response

Responses to Reviewer #2’s comments

Comment 1: 

Authors reviewed anticancer effects of cantharidin in various cancer cell lines, mainly relevant to therapeutic effects. This focus is not consistent with the title of this manuscript “Molecular Mechanism of Cantharidin Carcinopreventive Potency…”. Cancer therapy and cancer prevention are different concepts.

Response:

Thank you very much for your suggestion. The title has been changed to “Anticancer Attributes of Cantharidin: Involved Molecular Mechanism and Pathways”.

Comment 2:

As a natural product, cantharidin induces multiple cellular effects which heavily overlap among cancer cell lines. It may not be necessary to list every tested cell lines unless a disease status represented by certain cells is unique to the biological effects of cantharidin.

Response:

Thank you very much for your advice. 

This review summarizes the available data about “anticancer effects of Cantharidin on different cancer cell lines”, which easily informed the readers the anticancer effects of Cantharidin on their interested caner types and the underlying mechanisms. Although Cantharidin might exert similar effect on some certain cancer cell types, it functions differently by altering distinct signaling pathways in diverse cancer cell lines.

For example, Cantharidin induced apoptosis of human pancreatic cancer cell line through the extrinsic apoptosis pathway by elevating the expression level of TNF-α, TRAIL-1, TRAIL-2 [53,54], but promote this process of human lung cancer cell line A540 via the intrinsic pathway by reducing Bcl-2 translation [35].

The same to cell cycle arrest. Cantharidin arrested bladder carcinoma cells at G0/G1 phase by up-regulating p21 and p53 gene translation level and down-regulating Cyclin E and CDC25C [55], but arrested colon cancer cells in G2/M phase via halting CDK1 activity [37]. etc.

Taken together, it is necessary to list the reported tested cell lines by Cantharidin in this review for presenting a more integrated article and enlarging the readership, to the best of author’s knowledge.

Comment 3:

Several sentences or sub-conclusions are misleading. Here are a few examples:

    a.“there is a great need to investigate naturally occurring bioactive compounds to fight this fatal disorder as natural toxins possess less pernicious effects.” Natural toxins could be highly detrimental to normal tissue. Authors may fit cantharidin with the concepts of modern drug discovery, emphasizing how the compound may  target the drivers of certain indication.     

    b.“Revesrsitol (I think the authors mean Resveratrol), another natural phenolic compound in red grapes and berries can be utilized to cure inflammatory bowel disease”. The compound was studied for this disease but can not provide a “cure”.  

    c.“… but its anticancer potential can’t be abandoned mainly due to its PP2A inhibiting ability.” Many  activities are reported, PP2A inhibition is just one of them.

Response:

    a.As suggested, the sentence has been revised to “Considering these dilemmas there is a great need to investigate naturally occurring bioactive compounds to fight this fatal disorder as natural toxins possess certain therapeutic effect on various diseases and are the valuable repository for modern drug discovery.”(page 2 line 46-47).

    b.Sorry for the typing mistake (Revesrsitol), it has been replaced with Resveratrol in the revised manuscript. Furthermore, the word “cure” has been changed with “improve prognosis in patients with inflammatory bowel disease” (page 2 line 49).

    c.Thanks for identification. To the best of author’s knowledge PP2A inhibition is the main mechanism of Cantharidin anticancer activity that’s why we included only this. But this sentence have been rephrased as “anticancer potential can’t be abandoned as it possess many pertinent anticancer effects on cancereous cells like PP1 and PP2A inhibition, apoptosis induction and alteration of protein synthesis” (page 2 line 75).

Comment 4:

Analogues of cantharidin could be compared in much detail, including chemical structures, therapeutic and toxic profiles, etc.

Response:

A figure for Cantharidin derivatives has been provided in the revised manuscript (Fig 2). This review focuses on the available data about “anticancer attributes of Cantharidin” only, but not its derivatives. Although the structures and anti-cancer effects of Cantharidin derivatives are important for design of new chemical molecules, which could be discussed in another manuscript.

Comment 5:

Figure 3. It is not clear if the figure is to describe the effects of cantharidin in cytoplasm and in nucleus, as cell membrane and nucleus were drawn. However, a lot of affected proteins in the cytosolic area of the picture exert their functions in the nucleus.

Response:

This picture has been drawn to describe which proteins or pathways Catharidin affect in nucleus or cytoplasm. According to suggestion the picture has been modified in the revised version of manuscript with correct location of affected proteins in nucleus and cytoplasm (Figure 4).

Comment 6:

Many misspellings throughout the manuscript need to be corrected.

Response:

Thanks for your indication. The manuscript has been reviewed carefully for misspellings and all possible misspellings have been corrected as;

    (Abstract line 14 terpinoid to terpenoid), (Abstract line 23 carcinoprevantive to carcinopreventive), (page 2 line 48 revesrsitol to resveratrol), (page 2 line 68 coupulation to copulation), (page 2 line 70 Antractic to Antarctic), (page 3 line 90 venomous warms to venomous worms), (page 3 line 85 scarfoderma to scrofuloderma), (page 4 line 120 programed to programmed), (page 9 line 271 metteloproteinases to metalloproteinases), (page 11 line 344 BLAB/c to BALB/c mice),  (page 11 line 371 indudtry to industry), (page 12 line 377 phosphoraylation to phosphorylation), (page 12 line 381 pharmacotheraputics to phracotherapeutics), (page 12 line 402 carcinoprevantive to carcinopreventive).

Reviewer 3 Report

The authors summarized the anticancer effect of Cantharidin (CTD) and its molecular mechanism in this review. The appropriate anticancer potency of CTD can be used to develop effective anti-carcinogenic drugs. The anticancer action of the various mechanisms of CTD is exciting.

I point out some minor revisions.

  • In Fig2, showing the carcinopreventive profile of CTD in a pie chart is not very effective. The proportion of each cancer looks the same.
  • The author has over-described the general mechanism of apoptosis. Readers are most interested in the apoptosis induction mechanism of CTD.
  • In Table1, ”↓ BCR-ABL protein expression “  Please rewrite ↓ as a phrase.
  • In Fig.3, I recommended that the molecules present in the membrane be arranged on a membrane.

Author Response

Responses to Reviewer #3’s comments

Comment 1:

In Fig2, showing the carcinopreventive profile of CTD in a pie chart is not very effective. The proportion of each cancer looks the same.

Response:

Actually this figure has been drawn just to show types of cancers that Cantharidin can prevent along with names of cell lines on which Cantharidin activity has been studied. It has not been drawn to show Cantharidin efficacy proportion on different type of cancers.

Comment 2:

The author has over-described the general mechanism of apoptosis. Readers are most interested in the apoptosis induction mechanism of CTD.

Response:

As suggested, the section of the general mechanism of apoptosis has been simplified. (page 4 line 136-138), (page 5 lines 150, 153, 182).

Comment 3:

In Table1, ”↓ BCR-ABL protein expression “  Please rewrite ↓ as a phrase.

Response:

Thanks for your suggestion. ↓ has been changed to phrase “down-regulation” (Table 1 page 7).

Comment 4:

In Fig.3, I recommended that the molecules present in the membrane be arranged on a membrane.

Response:

This picture has been drawn to describe which proteins or pathways Catharidin affect in nucleus or cytoplasm. According to suggestion the picture has been modified in the revised version of manuscript with correct location of affected proteins in membrane, nucleus and cytoplasm (Figure 4).

Reviewer 4 Report

Title: Molecular Mechanism of Cantharidin  Carcinopreventive Potency in Various Cancer Cell Lines

In this mini-review the authors summarize recent information about CTD carcinopreventive potential and underlying molecular mechanisms. The pertinent anticancer strength of CTD could be employed to develop an effective anticarcinogenic drug.

The topic is interesting and timely, though minor revisions could improve the article for a possible publication in Molecules Journal.

The manuscript also contains several grammatical and spelling errors.

Does Cantharidin exert effects on angiogenesis? Could the authors add also this information.

It looks like the manuscript has not been extensively revised by authors, there are many typos and spelling errors in the manuscript.

abstract line 21 "carcinoprevantine " instead of " carcinopreventive "

page 2  line 46 "Revesrsitol" instead of "Resveratrol"

page 4 line 117 "programed" instead of "programmed"

page 5 line 170 "CTD exert" instead of "CTD exerts"

page 10 line 264 "metteloproteinases" instead of "matalloproteinases"

page 10 line 267 "CTD remarkably exert" instead of " CTD remarkably exerts "

page 11 line 336 "BLAB/c" instead of "BALB/c"

page 11 line 363 "indudtry" instead of "industry"

page 12 line 393 "carcinoprevantine " instead of " carcinopreventive "

Author Response

Responses to Reviewer #4’s comments

Comment 1: 

The manuscript also contains several grammatical and spelling errors. It looks like the manuscript has not been extensively revised by authors, there are many typos and spelling errors in the manuscript.

Response:

Thanks for your recognition. Sorry for these mistakes. The manuscript has been reviewed carefully for grammatical and spelling errors and several mistakes have been corrected as:

 (Abstract line 14 terpinoid to terpenoid), (Abstract line 23 carcinoprevantive to carcinopreventive), (page 2 line 48 revesrsitol to resveratrol), (page 2 line 68 coupulation to copulation), (page 2 line 70 Antractic to Antarctic), (page 3 line 85 scarfoderma to scrofuloderma), (page 4 line 120 programed to programmed), (page 9 line 271 metteloproteinases to metalloproteinases), (page 11 line 344 BLAB/c to BALB/c mice), (page 11 line 371 indudtry to industry), (page 12 line 377 phosphoraylation to phosphorylation), (page 12 line 381 pharmacotheraputics to phracotherapeutics), (page 12 line 402 carcinoprevantive to carcinopreventive), (Page 5 line 175 CTD exert to CTD exerts), (Page 10 line 274 CTD remarkably avert to averts).

Comment 2:

Does Cantharidin exert effects on angiogenesis? Could the authors add also this information.

Response:

Yes, we tried to summarize effect of Cantharidin on angiogenesis. As “Furthermore, reduction in the number of completely formed tubes, and vascular sprouting density and length indicated inhibition of angiogenesis in these mice” (page 12 line 336-337).

Reviewer 5 Report

Naz et al. reviewed the literature on the chemopreventive activity of CTD. They emphasize that CTD can kill of the Imatinib-resistant CML cell line and difficult to treat, triple-negative breast cancer cells. One of the advantages of this chemical is the effect on many signal tranduction pathways in cancer cells. The publication is interesting but needs improvement.

Introduction, p. 2, line 46; do the authors mean resveratrol?

Regarding chapter 4.3 .: Authors should refer to the possble impact of CTD on DNA of normal cells. This problem definitely requires further in vitro and in vivo testing in normal cells, especially myeloid cells.

Fig 3 legend, the authors should explain the meaning of all characters - explain the meaning of a blue triangular arrow or a rounded arrow.

Authors should improve typographical errors.

Author Response

Responses to Reviewer #5’s comments

Comment 1:

Introduction, p. 2, line 46; do the authors mean resveratrol?

Response:

Thanks for your recognition. Yes, it means resveratrol and it has been changed in the revised version.

Comment 2: 

Regarding chapter 4.3 .: Authors should refer to the possible impact of CTD on DNA of normal cells. This problem definitely requires further in vitro and in vivo testing in normal cells, especially myeloid cells.

Response:

To the best of author’s knowledge, the effect of this compound on normal DNA has not been found in the literature. Since, this manuscript is reviewing the available literature of “different anticancer attributes of Cantharidin”. Therefore, to discuss the effect of this compound on normal DNA by conducting experiment is not in the scope of this review.

Comment 3: 

Fig 3 legend, the authors should explain the meaning of all characters - explain the meaning of a blue triangular arrow or a rounded arrow.

Response:

Thank you for your affirmation. Figure 3 has been revised and meanings of all possible characters have been explained (Figure 4).

Comment 4: Authors should improve typographical errors.

Response:

Sorry for these mistakes. The whole manuscript has been reviewed carefully and all possible mistakes have been corrected as:

    (Abstract line 14 terpinoid to terpenoid), (Abstract line 23 carcinoprevantive to carcinopreventive), (page 2 line 48 revesrsitol to resveratrol), (page 2 line 68 coupulation to copulation), (page 2 line 70 Antractic to Antarctic), (page 3 line 85 scarfoderma to scrofuloderma), (page 4 line 120 programed to programmed), (page 9 line 271 metteloproteinases to metalloproteinases), (page 11 line 344 BLAB/c to BALB/c mice), (page 11 line 371 indudtry to industry), (page 12 line 377 phosphoraylation to phosphorylation), (page 12 line 381 pharmacotheraputics to phracotherapeutics), (page 12 line 402 carcinoprevantive to carcinopreventive), (Page 5 line 175 CTD exert to CTD exerts), (Page 10 line 274 CTD remarkably avert to averts).

Round 2

Reviewer 2 Report

The authors made extensive corrections and revisions on the previous version. The reviewer appreciate these changes.

Author Response

Responses to Reviewer #2’s comments

Comment 1: 

Extensive editing of English language and style required 

Response:

Thanks for your suggestion. The manuscript has been revised carefully to correct all possible English language errors.

-First letters of the key words have been changed to the lower case.

-Corrected use of articles (remove the from the line “hallmarks of cancer”, page 13 line 326), (inserted ‘a’ in the line “In this study, cisplatin was used as a positive control”, page 13 line 353),

-Corrected spelling (proangeogenic to proangiogenic, page 13 line 350),

-Corrected punctuation (comma) (page 1 line 29, 30), (page 4 line 103), (page 9 line 151), (page 10 line 188), (page 10 line 27), (page 12 line270), (page 13 line 340), (conclusion line 432, 433, 437), (page 12 line 262), (page 14 line 352 and 363).